# The Role of the Law in Prompting Environmental Stewardship for Farms Located Near Phosphate Mines

**Abdullah Banikhalid and Michel Rahbeh ***

Department of Land, Water, and Environment, The University of Jordan, Amman 11942, Jordan; abanikhalid@yahoo.com
* Correspondence: m.rahbeh@ju.edu.jo

**Abstract:** The effectiveness of environmental laws depends on several factors, including the cooperation between the stakeholders, compliance, and implementation. This research investigated the effectiveness of the Jordanian environmental laws in protecting agricultural lands near phosphate mines that are operated by the Jordanian Phosphate Mining Corporation (JPMC). The two other stakeholders involved are the Ministry of Environment (MOENV) and the farmers. The evaluation of the effectiveness of the environmental law was based on a hypothetical model that considers consecutive relations between awareness, commitment, and compliance. A second model was based on monitoring and enforcement as catalysts to the awareness and commitment that lead to compliance. The research problem was addressed using three questionnaires based on a 5-point Likert scale. The results showed modest compliance by the farmers and the absence of a correlation between awareness and commitment, as well as a lack of monitoring by the MOENV. The consecutive model can explain the compliance of the JPMC, since awareness, commitment, and compliance were well correlated despite the lack of managerial involvement by the JPMC in enhancing environmental awareness. It is recommended that more resources be allocated to increase the monitoring activities by the MOENV and study the social and economic factors influencing farmers' compliance.

**Keywords:** environmental law; mining industry; monitoring; awareness; commitment; compliance

## 1. Introduction

### 1.1. Background

Phosphorus is an essential plant nutrient; therefore, phosphorus fertilizers are of paramount importance to food production. Phosphorus is also used in the manufacture of cement and concrete [1].

Phosphate mining from phosphate rocks is the main source of phosphorus. The most phosphate-producing countries in the world are Morocco, the United States, and South Africa, which own 21, 4.2, and 2.5 billion tons of the global reserve, respectively [2]. The amount of phosphorus that is mined every year is 176 million tons, and the demand for phosphate fertilizers is expected to increase in the future [3]. Employment is another economic benefit of phosphate mining. For example, in Brazil, more than 198,000 individuals were employed at a cost of approximately USD 14 billion in the first half of 2014 [2].

Jordan's mineral mining industry is one of the country's major industries since it has abundant supplies of several minerals, including cement, phosphate, and calcium carbonate. Many local firms have partnered with global corporations to explore and extract these minerals, which is a major source of investment for Jordan. Phosphorus made up 8.5%, potassium 12%, acids 2.6%, fat 0.6%, bromine 2.4%, and total fertilizer exports 9.2% of the local mineral exports in 2009 [4].

Unfortunately, the economic benefits of the phosphate industry come with a hefty price on the environment. As waste is disposed of in various environments, phosphate mining releases large amounts of minerals into the environment, such as phosphate gypsum,

which consists of calcium sulfate and other salts [5]. Hydrological studies indicated that phosphate mining decreased runoff and runoff peak and increased total nitrogen (TN), soluble phosphorus (SP), and total phosphorus (TP) [6,7]. The coral reef in the vicinity of a primary phosphate storage facility was adversely affected by the deposition of sediment rich in phosphate, hindering the long-term viability of the coral reef ecosystem [8]. Watersheds with active reclaimed phosphate mining areas are a major source of P in stream water that may alter the P:N ratio and lead to eutrophication [9]. Furthermore, it has been found that phosphate mining releases heavy metals, such as cadmium and arsenic, which cause soil, air, and water pollution [10–13].

Environmental legislation is interconnected with basic human rights, specifically the right to a healthy environment [14]. It should be noted that the majority of environmental legislation is fragmented and divided into several laws. For example, in the United States, the environmental laws include the Clean Air Act (CAA); Clean Water Act (CWA); Resource Environmental Conservation and Recovery Act (RCRA); Comprehensive Environmental Response, Compensation, and Liability Act (CERCLA); and Endangered Species Act (ESA) [15]. Within the European Union, guidance in the form of a uniform soil directive does not exist; hence, member states enact their own legislation governing historic soil contamination [16].

In Jordan, the environmental legislation consists of several passages from different laws which do not synchronize with each other. In 2003, an environmental law was issued which was subsequently ratified in 2006. It provided the necessary legislative framework to issue numerous detailed regulations pertaining to environmental protection, including the mandate for the Ministry of Environment (MOENV) to protect and maintain the environment. Furthermore, these laws were revised in 2017. The new amendments mandated the creation of an Environmental Protection Fund, which would be used to fund environmental preservation [17].

However, the effectiveness the Jordanian environmental laws are yet to be examined, especially when it comes to the serious environmental hazards posed by the mining industry. In particular, Jordan's environmental circumstances are deteriorating remarkably year after year, with occasional hotspots that required considerable attention from governmental institutions and numerous constituents to address and devise workable remedies. Although urbanization's encroachment on green spaces and the decline of agricultural areas may rank among the most significant unresolved environmental issues, the pressure on environmental advocates and the variety of hazards they face motivates them to make every effort to protect the environment. Furthermore, environmental awareness is rapidly growing in Jordan, since upholding a safe and secure environment is now seen as a human right. The plans and strategies set in place to address the current difficulties are based on a certain view from a specific angle, but the Kingdom faces other environmental concerns that are not given enough attention, particularly with the frequent changes in ministers. The foundations of sustainable development are threatened by the prohibitive human, economic, and social costs associated with environmental deterioration. One of the largest environmental issues facing Jordan is waste, as the country produces 1.7 million tons of solid waste annually at a rate of 3850 tons per day. Of this, 60% is organic waste that is dumped in one of the country's 21 landfills, which are regarded as unsanitary dumps. Given that most of Jordan's territory is either semi-arid or dry, the desertification phenomenon is also one of the most significant and dangerous environmental issues posing a threat to agricultural fields. Jordan is confronted with several environmental obstacles as well, such as unique problems pertaining to certain polluted regions that are out of date and ignored by governments, thus aggravating the state of the environment. One of the most significant issues facing the environmental reality of the Kingdom is the existence of environmentally hot areas that require drastic treatment.

The phosphate mining industry is one of the main environmental concerns in Jordan. There are four main mining locations (Figure 1). The Eshidiyah mine is located 125 km northeast of the gulf of Aqaba and produces 5 million tons annually. The Russifa mine,

which is the oldest, is located 12 km northeast of Amman the capital of Jordan, is regarded as one of the aforementioned environmental hotspots due to the presence of numerous factories and craft areas that have contributed to creating a harsh environment in that densely populated area [18]. This mine ceased mining due to its negative effects on the surrounding environment and population density. Wadi Al-Abiad mine and Al Hasa mine are located 115 km south of Amman. These mines are operated by the Phosphate Mines Company (JPMC). The JPMC is regarded as one of the major contributors to environmental pollution [19]. The operation of these mines poses serious hazards to the environment as the concentration of uranium and cadmium in the soil is higher than what is allowed for cultivation. The phosphate deposits in Jordan are noticeably richer in uranium than those in other parts of the globe [20]. In addition, the water used in the JPMC mines is disposed of in the desert without treatment, which negatively impacts the groundwater [21]. Additionally, the water effluent from the mines is used to irrigate the agricultural lands in the vicinity of the Wadi Al-Abiad mine and Al Hasa mine.

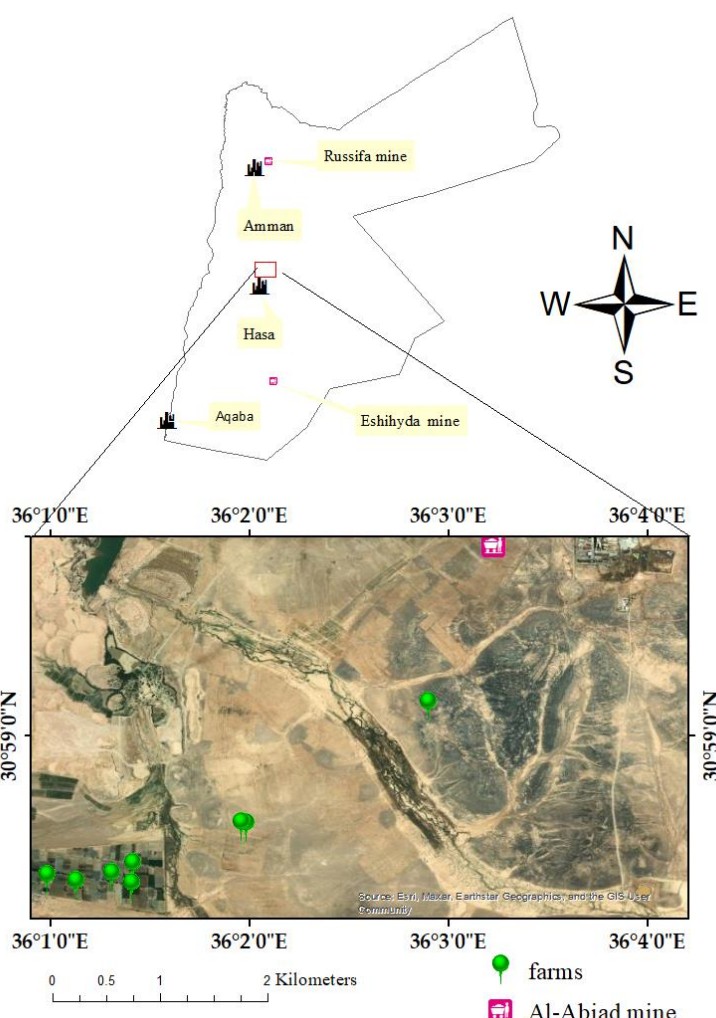

**Figure 1.** The locations of the phosphate mines of the JPMC and the farms near Al-Abiad mine.

In view of the environmental challenges Jordan is facing and the serious hazards of the phosphate mining industry, it is important to evaluate the compliance with the Jordanian environmental laws as a means of protecting the agricultural land near these phosphate mines from the hazards associated with both agricultural practice and mining operations.

### 1.2. Compliance with the Environmental Laws

Compliance with environmental laws can be inferred through three motivations: calculated, normative and social, in addition to knowledge and awareness [22].

Calculated motivation is based on the desire of the regulated entities to increase the benefits of compliance by averting the costly penalties and fines [22] or reap the incentives provided by the regulating agencies. For example, agriculture certification, which allows farmers to sell their products at higher prices in the premium markets, enhanced compliance with environmental legislation [23]. In Serbia, farmers were willing to adopt agri-environmental measures in return for long-term economic benefits [24]. Neves et al. [25] found that penalizing polluting activities in the form of taxes and reinforcing clean activities by subsidies and financial benefits were effective in reducing $CO_2$ emissions in European Union countries. In all cases, calculated motivation cannot be achieved without effective monitoring and enforcement by the government [26]. A review of environmental law enforcement in the United States concluded that enforcement and monitoring not only reduced violations but also emissions [27].

Normative motivation is the commitment of regulated entities to comply with the environmental laws, driven by either moral principles or an appreciation of the reasonableness and the values of the laws [22]. Li et al. [28] suggested that successful implementation of the laws is linked to adopting legal regulations compatible with the moral values of the regulated society. It could also be linked to environmental literacy or education, which is a combination of awareness, knowledge, involvement, attitude, and behavior [29]. Using the theory of planned behavior, Su et al. [30] showed that attitudes, personal factors, and behaviors positively influenced farmers' intention to adopt environmentally friendly practices and join activist organizations. Ham et al. [31] highlighted the importance of a positive attitude or reaction toward environmental issues, which stems from either concerns about the environment or awareness of the vital function it provides for maintaining and improving the quality of life. Furthermore, increased public awareness of environmental pollution also increases the expectations concerning environmental governing [32], and a lack of public awareness and knowledge about environmental issues limits people's willingness to participate [33]. Onyando et al. [34] found that low awareness may hinder the safe use of pesticides by farmers. Vapa Tankosić et al. [24], found that farmers were more willing to accept the AEM if they felt that preservation of the environment was essential for protecting the resources for future generations. Also, the adoption of AEM by farmers can be enhanced by increasing training and environmental education that demonstrate the environmental benefits. Tian et al. [35] showed that environmental values and awareness correlated positively with farmers adopting practices that prevent excessive fertilization. Niu et al. [36] devised a model to explain the influence of environmental education on environment governing. The results showed that environmental education does not directly affect environmental governing, but it does have an effect on public participation, which in turn can lead to improvements in environmental governing.

Social motivation involves the desire to earn the approval of the community and representatives of the enforcing agencies, such as inspectors and other regulated entities [22]. It has been shown that communication of information between farmers (social networking) moderated the influence of awareness and environmental values and enhanced the adoption of environmentally friendly practices [35]. A major mining company in Ghana, guided by its corporate image, voluntarily adhered to the environmental regulations to the satisfaction of the local community and auditors [37]. Flores et al. [38] found that pro-social variables have a greater impact on influencing farmers to adopt land conservation practices than economic incentives. The compliance of Chinese corporations with the environmental law was enhanced by soft law, which is defined as a non-legally binding set of ethical considerations, recommendation by the government, and social responsibility toward the society [39].

Knowledge and awareness are an essential element of compliance; simply, the regulated individual or entity will not comply if they are unaware or have insufficient knowl-

edge about the regulations. In China, farmers' legal cognition, described as knowledge and legal familiarity, positively moderated their environmentally friendly intentions, which was inferred by the effect of increasing legal cognition on enhancing farmers' appreciation of the rules and environmental regulation and encouraged by the costs and incentives of the law to abide by the environmental law [30]. An investigation of Danish farmers' compliance with environmental regulations showed that the framers' compliance was critically influenced by their awareness of the rules and regulations. Compliance was also enhanced by normative motivation and calculated motivation provided that law enforcement is formal but without coercion [22]. Alotaibi et al. [40] suggested that farmers' awareness about the environmental laws was linked with their awareness of the environmental hazards related to the use of agrochemicals.

### 1.3. Theoretical Models

Theoretical models have been used to identify the independent variables that influence adherence to beneficial environmental practices [24], compliance with the environmental law [22], environmental governess [36], and stewardship and responsible environmental behavior [30]. The models can be straightforward evaluations of the independent variables' connections with the dependent variables. Winter and May [22] examined the direct effects of calculated, normative, and social motivations on compliance with agro-environmental rules, and Vapa et al. [24] investigated the variables that influence the improvement of the environment. It is also possible to consider variables that moderate or enhance the influence of the independent variables. Su et al. [30] considered environmental law cognition to moderate the relationship between environmental intention and responsible environmental behavior. Niu et al. [36] considered a direct connection between environmental education and environmental governess, also the willingness to participate and ways to participate were considered as moderating variables. Floress et al. [38] hypothesized that stewardship attitudes moderated the influence of farmers' awareness and business attitudes on willingness to improve water quality. However, the existing models do not consider normative variables as moderating variables toward compliance. Therefore, we envisaged two theoretical models that consider commitment and awareness either as independent or moderating variables toward compliance. The rationale and development of the theoretical models are further discussed in Section 2.3.

### 1.4. The Significance and Objectives

Enforcement is the first intuition for achieving compliance with the law [26,27]. However, the review presented in the previous sections indicated that law enforcement alone may be insufficient to achieve effective compliance with the environmental rules and regulations. Instead, it should be combined with awareness about the laws and environmental problems and commitment, which includes a variety of social and moral variables [24,30,34,35]. However, previous research focused on normative motivations separate from law enforcement and without sufficiently addressing the question of the effectiveness of the environmental laws. Interestingly, the results obtained by Su et al. [30] suggest that the farmers' intentions and behavior were moderated by the knowledge of the consequences of not complying, and abiding by the law was incentivized, provided that the environmental law was diligently implemented by the government. Thus, two gaps can be identified. The first gap is that, despite these valuable insights, it is still unclear how these motivations interact with each other, whether enforcement is the main catalyst for compliance or mediated by awareness and commitment, and whether compliance stems from normative variables, such as moral obligations to the environment and society. The second gap highlights the lack of studies on the motivations of the Jordanian stakeholders that lead to compliance with the Jordanian environmental laws.

The objective of this research was to address these two gaps by investigating the motivations and interactions between them that lead to compliance by the JPMC and farmers of the agricultural land near the phosphate mines. This study aids in explaining

the role of the Jordanian environmental laws in promoting environmental stewardship and the role of law enforcement in improving the effectiveness of the environmental laws and protection. From the global perspective, it not only adds to the topic of compliance with environmental laws and regulations, it also explains the interconnection between calculated and normative motivations, and how law enforcement affects commitment and awareness in order to ensure the effectiveness of the environmental law.

## 2. Materials and Methods

### 2.1. The Variables for Compliance

The literature review on compliance with environmental laws (Section 1.2) revealed that the variables can be classified into three major motivation categories [22]. The first category is "calculated motivation", which directs the regulated entities to avoid the unnecessary cost of enforcement in the form of fines and penalties by adhering to the requirements of the laws. However, enforcement is meaningless without awareness of the laws and monitoring the violations [26,27]. Thus, the variables relevant to calculated motivation are "monitoring", "enforcement" and "awareness", which conveys general knowledge about the laws. The second category is "normative motivation", which includes commitments to the environment as a matter of principle and wrapped up with moral values. It appears that environmental education plays an important role in promoting "normative motivation" and also overlaps with the third category, "social motivation"; although this is driven by the image of the corporate entities, the regulated entities may also be consciousness of their environmental obligations to the local community [28,37,38]. Thus, by defining "commitment" as a positive attitude and actions not necessarily mandated by the laws or enforced by the means of fines or penalties, it can refer to both normative and social motivations.

### 2.2. Stakeholders

Previous research on compliance with environmental laws (Section 1.2) focuses on the regulators, such as law enforcement agencies, and the regulated entities, such as the farmers and mining and industrial enterprises. Other groups of stakeholders may also include the general public and environmental advocacy groups. Describing the selection of stakeholders relevant to a certain issue or project as an "art" or "science", Colvin et al. [41] proposed that the identification of stakeholders may be based on intuition as well as more objective approaches, such as the geographic footprint, interests and influence. For this study, the enforcement agency is the MOENV, which is represented by its employees. They are the first group of stakeholders because of their interests in implementing laws, and they can influence the compliance of the regulated stakeholders, including the farmers of the agriculture land located near the JPMC and the JPMC. Both the JMPC and farmers fit the criteria of the geographical study area, and they directly influence the environmental drivers defined as the underlying elements or forces of the natural world that are protected from harmful human activity through the establishment of environmental regulations [25]. The JPMC mines pollute the environment by releasing heavy metals and emissions. Farmers also contribute to the pollution through the use of agrochemicals and the effluent from the JPMC mines. Additionally, the farmers near the JPMC mines are beneficiaries of the environmental laws, since among the objectives of the environmental laws are the protection of soil and land resources, which directly aid in the sustainability of agriculture.

### 2.3. Two Theoretical Models

The smooth and successful implementation of the environmental laws depends on their positive reception by each of these groups and a healthy interaction between the stakeholders. Compliance may define the interaction between MOENV and the other two stakeholders. To build a theoretical model some support should be provided for the connections between variables in the models [38]. For the first theoretical model (model 1) it was hypothesized that awareness, commitment and compliance are consecutively related

to each other accordingly (Figure 2). As regards the connection between awareness and commitment, it was found that awareness about either the environmental laws or the environmental issues can enhance the commitment of the regulated stakeholders [29,33,35]. This connection also finds support from the results obtained by Vapa Tankosić et al. [24], which indicated that education and training, enhance farmers' commitment to adopt AEM. As for the connection between commitment and compliance, previous results showed stewardship attitudes were positively related to willingness to take action to protect the environment [38], and commitment led a mining corporation to comply with the environmental law [37]. Furthermore, Winter et al. [22] recommended the inclusion of non-deterrent measures to enhance compliance with environmental laws. Thus, providing sufficient support for the connection between commitment and compliance.

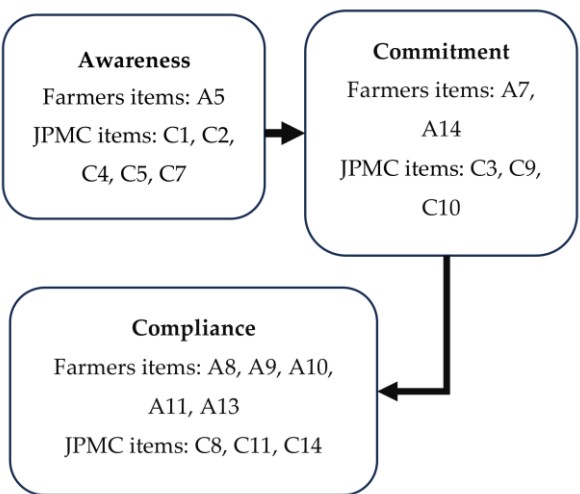

**Figure 2.** The hypothetical model 1 describes a consecutive relationship between awareness, commitment and compliance. The questionnaires' items are specified for each component.

Alternatively, compliance with the environmental laws can be based on enforcement (model 2), as long as enforcement is preceded by effective monitoring and moderated awareness and commitment (Figure 3). The role of monitoring and enforcement for achieving compliance with environmental laws was confirmed in the EU [26] and the United States [27]. Furthermore, in Saudi Arabia, the enforcement of the environmental law was hindered by the lack of monitoring [40]. Thus, confirming the connection between monitoring and enforcement. The results of Altotaibi et al. [40] also indicated that enforcement affected farmers' legal and environmental awareness and that compliance was also associated with awareness. These findings support the connection between enforcement and awareness and the connection between awareness and compliance. Su et al. [30] also indicated the importance of legal cognition, preceded by the implementation of environmental regulations in moderating normative motivation, which provides sufficient support for the enforcement and commitment connection.

The research theories were investigated using three questionnaires, one for each stakeholder. They were constructed based on a five-point Likert scale (i.e., strongly disagree, neutral, agree, and strongly agree). The questions were classified into compliance (CP), awareness (A), commitment (CT) and law enforcement (LE).

The questionnaires were distributed to three groups of stakeholders. The first group comprised the employees of the Jordanian Ministry of Environment who have the mandate to protect the environment. The second group comprised the JPMC, which operates the phosphate mines. The third group comprised the farmers in the vicinity of the mines. The number of participants in each group exceeded 30, which was deemed statistically acceptable [42]. The study sample consisted of the following:

- The Jordanian Ministry of Environment (MOENV): the questionnaire consisted of 15 items and was distributed to 43 employees in the Jordanian Ministry of Environment from 25 January to 8 February 2023.
- The Jordan Phosphate Mines Company (JPMC): the questionnaire consisted of 17 items and was distributed to 36 workers of the Jordan Phosphate Mines Company (PCL) from 22 May through 25 May 2023.
- The beneficiaries of the Jordanian environmental laws (farmers): the questionnaire consisted of 20 items and was distributed to 35 farmers from 24 August to 27 August in 2023 during a field visit to the study area.

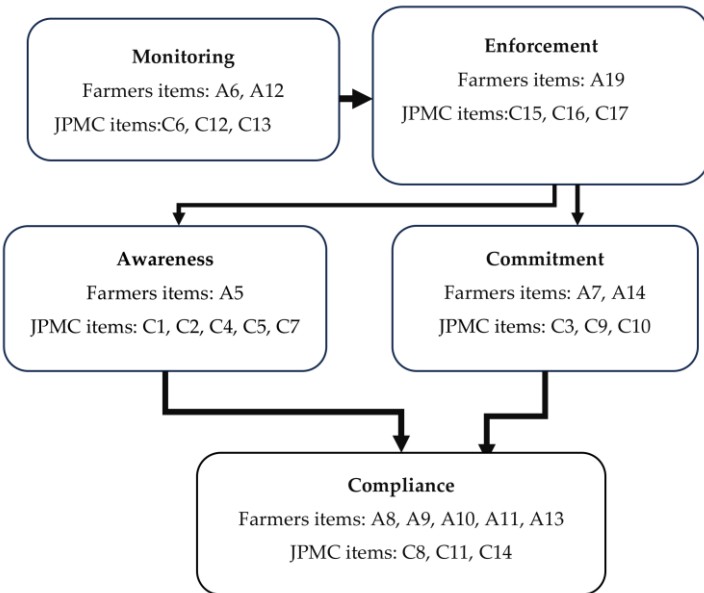

**Figure 3.** The hypothetical model 2 describes the relationship between awareness, commitment, and compliance. The model considers monitoring and enforcement as catalysts for awareness and commitment. The questionnaires items are specified for each component.

### 2.4. Statistical Methods

The questionnaire responses were analyzed using SPSS software (version 26). The outputs of the descriptive statistics included the mode, mean and frequencies. To ensure a scale that was free of errors and provided a consistent measurement across the different elements in a device [43], the Cronbach reliability test was used to evaluate the internal consistency of the questionnaire's items. In addition, Pearson's correlation ($r$), which provides the strength and direction of the bivariate linear relationships between variables [19,44], was used to study the relationship between any two questionnaire items. For the purpose of interpreting the Pearson correlation, $r$ values of $|0$ to $0.3|$, $|0.3$ to $0.5|$ and $|0.5$ to $1.0|$ were considered weak, moderate and strong, respectively. A significance Pearson correlation was indicated at $p < 0.05$.

## 3. Results

### 3.1. Reliability

The Cronbach reliability test required a reduction in the original number of items in the farmers' questionnaire to 12 items with a mean Cronbach's alpha of 0.867. The number of items of the MOENV and JPMC questionnaires remained the same with Cronbach's alpha values of 0.954 and 0.816, respectively.

*3.2. Descriptive Statistics*

3.2.1. The Farmers' Questionnaire

The descriptive statistics of the farmers' questionnaire (Table 1) highlighted the mean and mode values of the 5-point Likert scale as well as the frequencies of the responses (Figure 4). The mode values for the farmers' questionnaire were either four or two, indicating "agree" and "disagree" responses, respectively. The items associated with a mode of two returned mean Likert scale response values of between 2.71 and 3.0, while the "disagree" frequency was between 54.3 and 62.9%. As for items associated with a mode of four returned mean Likert scale response values of between 3.14 and 4.0, while the frequency for the "agree" response was between 40 and 68.6%. The farmers' responses showed compliance with the bounds of the environmental laws (item A10), following the regulations mandated such as obtaining the necessary permits (item A8), proper disposal of hazardous wastes (item A11), and disclosure of all the materials used (item A13). The farmers compliance with the environmental laws was confirmed by a strong Pearson's correlation coefficient [R] ($r > 0.5$) between all the compliance-related questions, namely A8, A10, A11, and A13 (Figure 5). The farmers also indicated that phosphate activity does not affect agriculture production (A18). However, their perception toward environmental laws was negative (A19), citing the lack of supervision by the MOENV for the disposal of hazardous materials (A12), although it was confirmed the continuous monitoring by the MOENV (A6). As for their own commitment to the laws, they showed a willingness to cooperate with the MOENV (A14), but monitoring of environmental risks was seen as a downside (A7). In fact, neither commitment question was significantly correlated; rather, A14 was only correlated with A12 ($r = 0.71$), which is one of the MOENV commitment questions, and A7 was strongly correlated with the compliance questions ($r > 0.5$). The farmers confirmed their own awareness (A5); however, A5 was only correlated with a commitment question (A6) ($r = 0.52$), and A6 was correlated with the law enforcement question (A19) (r = 0.64).

**Table 1.** Descriptive statistics from the 5-point Likert scale analysis of farmers.

|  | Classification | Items | Mode | Mean | Std. Error of Mean | Std. Dev. |
|---|---|---|---|---|---|---|
| A5 | Awareness | Always following the awareness bulletins issued by the Ministry of Environment | 4 | 3.20 | 0.158 | 0.933 |
| A6 | Commitment (MOENV) | There is continuous monitoring by the Ministry of Environment | 4 | 3.57 | 0.165 | 0.979 |
| A7 | Commitment | I monitor the environmental risks | 2 | 2.89 | 0.168 | 0.993 |
| A8 | Compliance | I obtain all necessary permits issued by the Ministry of Environment | 4 | 3.77 | 0.197 | 1.165 |
| A9 | Compliance | I am always ready for all inspection campaigns at any time | 4 | 3.71 | 0.145 | 0.860 |
| A10 | Compliance | I make sure that all my work is within the scope of the law and do not exceed it | 4 | 4.14 | 0.124 | 0.733 |

<div align="center">**Table 1.** *Cont.*</div>

| | Classification | Items | Mode | Mean | Std. Error of Mean | Std. Dev. |
|---|---|---|---|---|---|---|
| A11 | Compliance | I dispose of waste according to the methods announced by the Ministry of Environment | 4 | 3.71 | 0.156 | 0.926 |
| A12 | Commitment (MOENV) | The Ministry of Environment supervises materials and waste and methods of destroying and disposing of them | 2 | 2.71 | 0.162 | 0.957 |
| A13 | Compliance | I disclose all materials used in my work that affect the environment | 4 | 3.77 | 0.143 | 0.843 |
| A14 | Commitment | I cooperate with the Ministry of Environment to accomplish its tasks | 4 | 3.14 | 0.179 | 1.061 |
| A18 | Compliance (JPMC) | Phosphate mining activities affect agriculture and production | 2 | 3.00 | 0.217 | 1.283 |
| A19 | Law enforcement | Environmental law is effective in protecting agriculture | 2 | 2.97 | 0.194 | 1.150 |

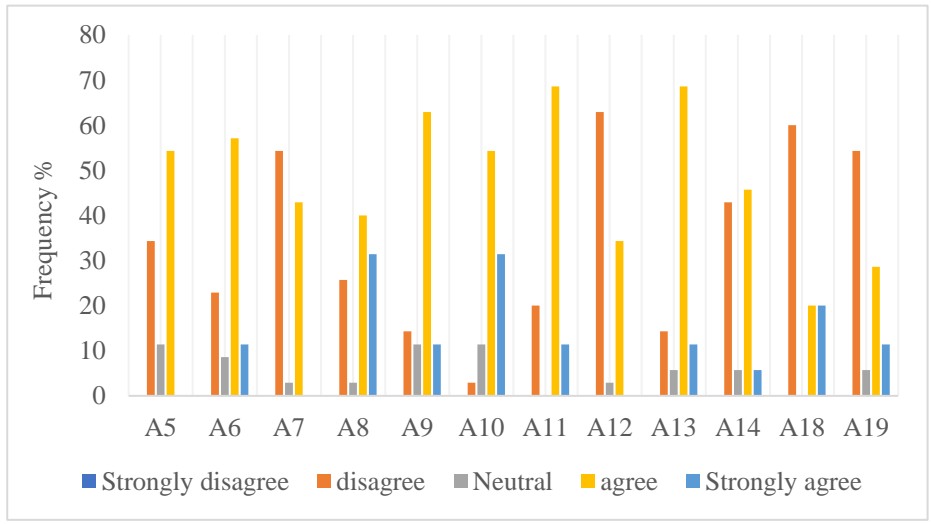

**Figure 4.** The frequencies of the farmers questionnaire.

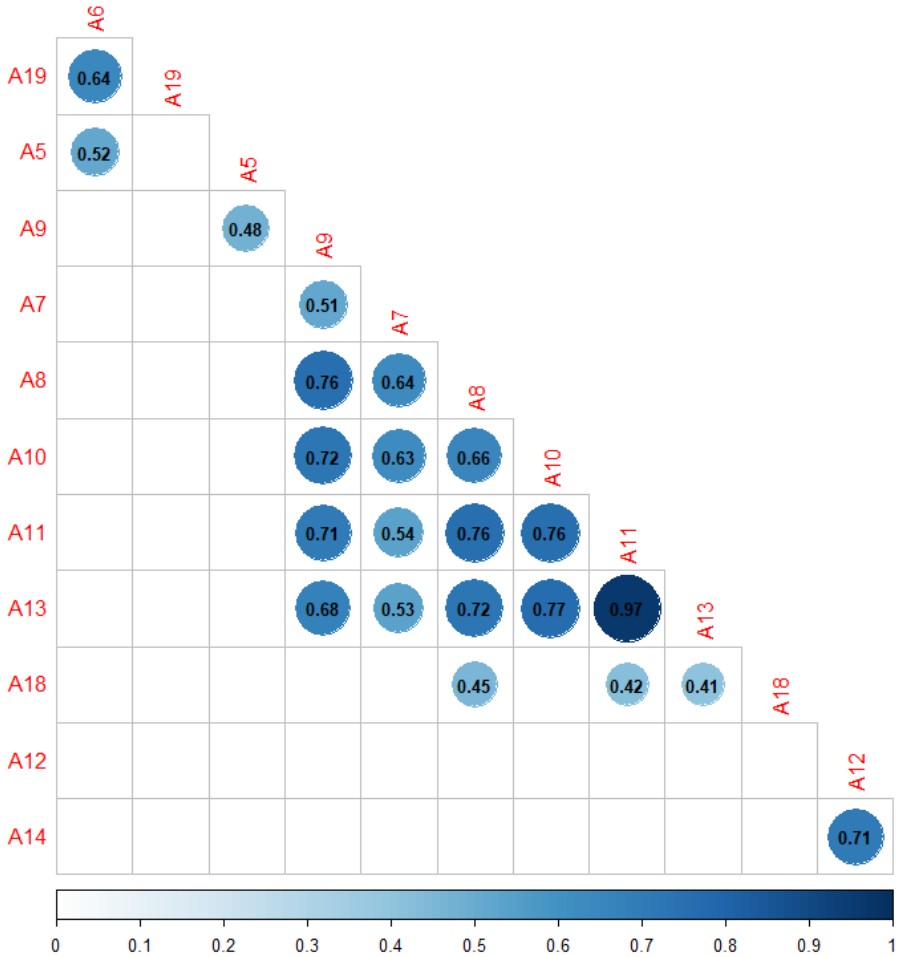

**Figure 5.** The Pearson correlation for the farmers questionnaire. The *r* values are shown only for statistically significant correlation at $p < 0.05$.

### 3.2.2. The JPMC Questionnaire

The descriptive statistics of the JPMC employees' questionnaire (Table 2) indicated the mean and mode values of the 5-point Likert scale and the frequencies of the responses (Figure 6).

**Table 2.** Descriptive statistics from the 5-point Likert scale analysis of the JPMC employee.

|  | Classification | Items | Mode | Mean | Std. Err Mean | Std. Dev. |
|---|---|---|---|---|---|---|
| C1 | Awareness | Awareness of the Jordanian environmental law (C1) | 4 | 3.29 | 0.162 | 0.957 |
| C2 | Awareness | Awareness of the procedures and reports issued by the Jordanian Environment Law (C2) | 4 | 3.34 | 0.183 | 1.083 |
| C3 | Commitment | The Ministry of Environment fully cooperates with The Jordan Phosphate Mines Company to implement the procedures of the law (C3) | 4 | 3.91 | 0.144 | 0.853 |

**Table 2.** *Cont.*

| | Classification | Items | Mode | Mean | Std. Err Mean | Std. Dev. |
|---|---|---|---|---|---|---|
| C4 | Awareness | The management organizes courses to increase knowledge about the role of the Ministry of Environment (C4) | 3 | 3.14 | 0.170 | 1.004 |
| C5 | Awareness | Mining phosphate Company publishes and manages awareness campaigns to highlight the importance of preserving the environment (C5) | 4 | 3.31 | 0.158 | 0.932 |
| C6 | Commitment (MOENV) | There is continuous monitoring by the Ministry of Environment (C6) | 4 | 3.77 | 0.164 | 0.973 |
| C7 | Awareness | There are awareness sessions for employees to introduce different environmental risks (C7) | 3 | 3.26 | 0.166 | 0.980 |
| C8 | Compliance | The company obtains all necessary permits issued by the Ministry of Environment (C8) | 4 | 4.09 | 0.126 | 0.742 |
| C9 | Commitment | The management sets plans in the follow-up process and conducts spot checks to ensure compliance with procedures and instructions (C9) | 4 | 3.46 | 0.144 | 0.852 |
| C10 | Commitment | The company cooperates by providing everything requested by the Ministry of Environment with all facilities to ensure the completion of their work (C10) | 4 | 4.06 | 0.116 | 0.684 |
| C11 | Compliance | The company disposes of waste according to the methods announced by the Ministry of Environment (C11) | 4 | 3.94 | 0.123 | 0.725 |
| C12 | Commitment (MOENV) | The Ministry of Environment supervises materials and waste and methods of destroying and disposing of them (C12) | 4 | 3.69 | 0.141 | 0.832 |
| C13 | Commitment (MOENV) | The Ministry of Environment supervises the monitoring and follow-up of excess emissions from mining areas (C13) | 4 | 3.86 | 0.137 | 0.810 |
| C14 | Compliance | The company is always ready for all inspection campaigns and at any time (C14) | 4 | 3.91 | 0.150 | 0.887 |
| C15 | Law enforcement | The penalties stipulated are sufficient to protect the environment (C15) | 3 | 3.40 | 0.160 | 0.946 |

**Table 2.** *Cont.*

| | Classification | Items | Mode | Mean | Std. Err Mean | Std. Dev. |
|---|---|---|---|---|---|---|
| C16 | Law enforcement | The Ministry of Environment takes appropriate legal measures to ensure the implementation of the law (C16) | 3 | 3.63 | 0.136 | 0.808 |
| C17 | Law enforcement | The Ministry of Environment deals firmly with violators of Jordanian environmental law (C17) | 3 | 3.49 | 0.155 | 0.919 |

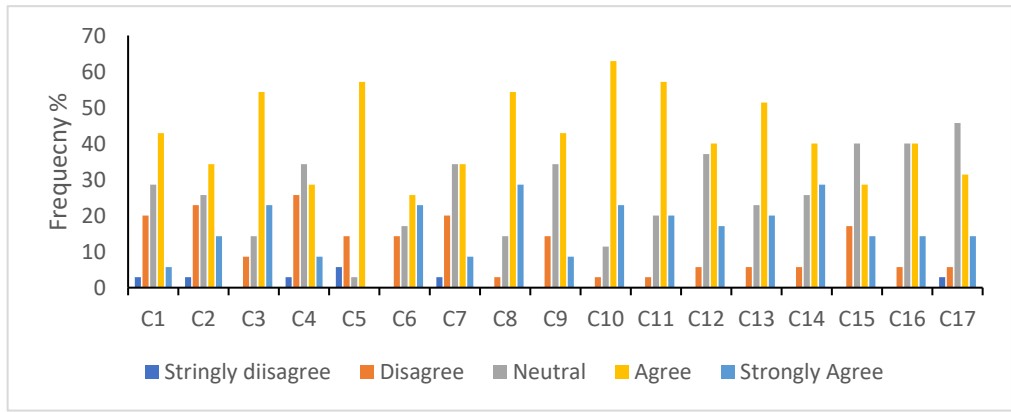

**Figure 6.** The frequencies of the JPMC employees' questionnaire.

The responses were split between mode values of four and three. Their responses were associated with a mode of four, with returned Likert scale values between 3.29 and 4.09, while for the responses associated with a mode of three, returned values were between 3.14 and 3.63. The responses were on the high side of the Likert scale, with prevailing "agree" and "disagree" responses. Six responses returned a frequency greater than 50% (Figure 6), comprising the JPMC employees' awareness (C5), compliance (C8 and C11), commitment (C3 and C10), and MOENV commitment (C12 and C13). A mode of three was associated with awareness (C4 and C7), which indicates a modest effort on the part of the JPMC managers to increase knowledge and awareness among its employees. Both the awareness questions were strongly correlated ($r = 0.77$). Additionally, a mode of 2 was associated with law effectiveness and enforcement (C15, C16 and C17), which indicates weak implementation of the laws, especially the inadequacy of the penalties prescribed by the laws for protecting the environment. These three questions were strongly correlated with each other ($r > 0.64$). Most of the responses were significantly correlated with each other at $p < 0.05$ (Figure 7). Thus, the commitment questions were strongly correlated with the awareness and compliance questions. However, it is also important to note that the MOENV commitment questions (C6, C12 and C13) were strongly correlated with the compliance questions (C8, C11 and C14), as indicated by the $r$ values between 0.6 and 0.88, respectively.

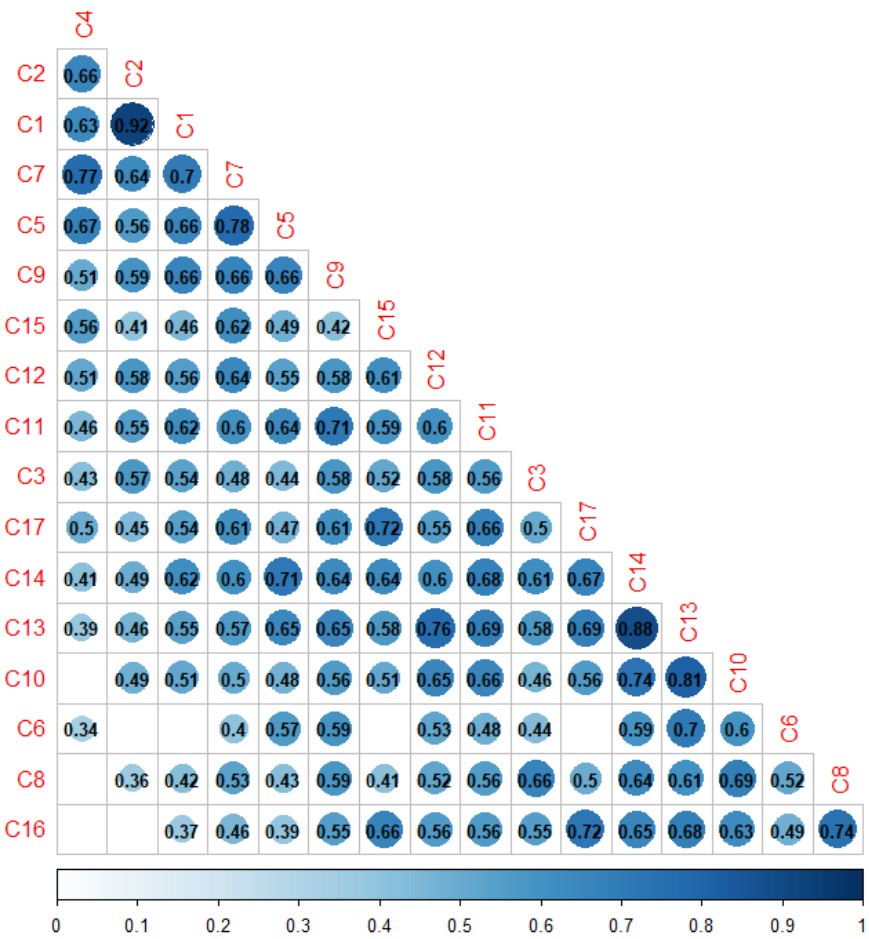

**Figure 7.** The Pearson correlation for the JPMC employees' questionnaire. The *r* values are shown only for statistically significant correlation at $p < 0.05$.

### 3.2.3. The MOENV Questionnaire

The descriptive statistics of the MOENV employees' questionnaire (Table 3) indicated the mean and mode values of the 5-point Likert scale and the frequencies of the responses (Figure 8). The responses were associated with three mode values of three, four and five. The mean Likert scale values associated with a mode of five were between 4.26 and 4.56. The modes of four and three were associated with values from 3.51 to 4.12 and 2.88 to 3.4, respectively. Similar to the JPMC employees' questionnaire, the responses of the MOENV employees were also on the high side of the Likert scale. The questions that returned low responses on the Likert scale (mode of three) were related to the farmers' compliance and commitment. The responses to E5 indicated a modest commitment by the farmers to the Jordanian environmental laws. Additionally, the responses to E5, E10, E11, and E12 indicated a low compliance since the farmers were not forthcoming with disclosing information and removing hazardous materials according to the approved procedures. The farmer compliance questions were moderately correlated to each other, as indicated by the *r* values between 0.34 and 0.44 (Figure 9). The strongest correlation was observed between E1 and E2 (*r* = 0.84), with both related to the awareness of the MOENV employees. The only correlation between the farmers' compliance and law enforcement was the between E11 and E8 (*r* = 0.4) and between E12 and E15 (*r* = 0.38). The awareness was moderately correlated with the commitment item, except for E2 and E3, in which the relation between the awareness of procedures and cooperation with the environmental police was tackled. The law enforcement questions were also moderately correlated with the commitment question; however, a strong correlation was observed between E3 and E8 (*r* = 0.52) as well as E2 and E12 (*r* = 0.61).

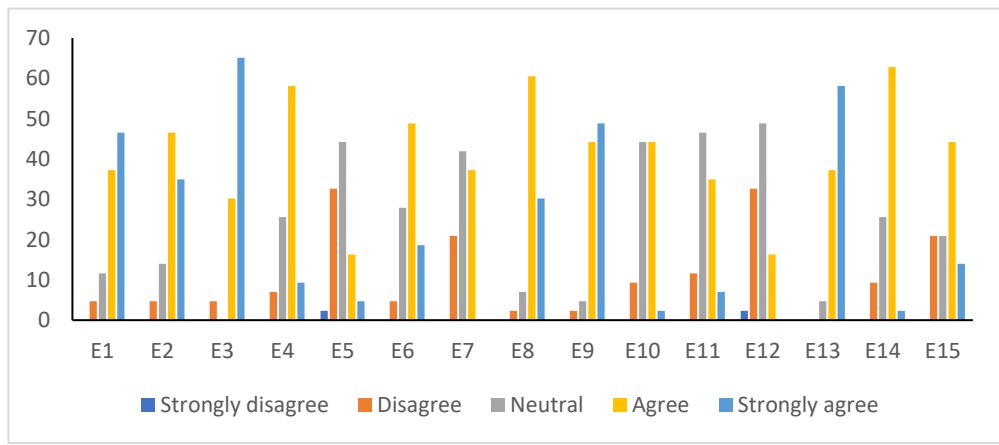

**Figure 8.** The frequencies of the MOENV employees' questionnaire.

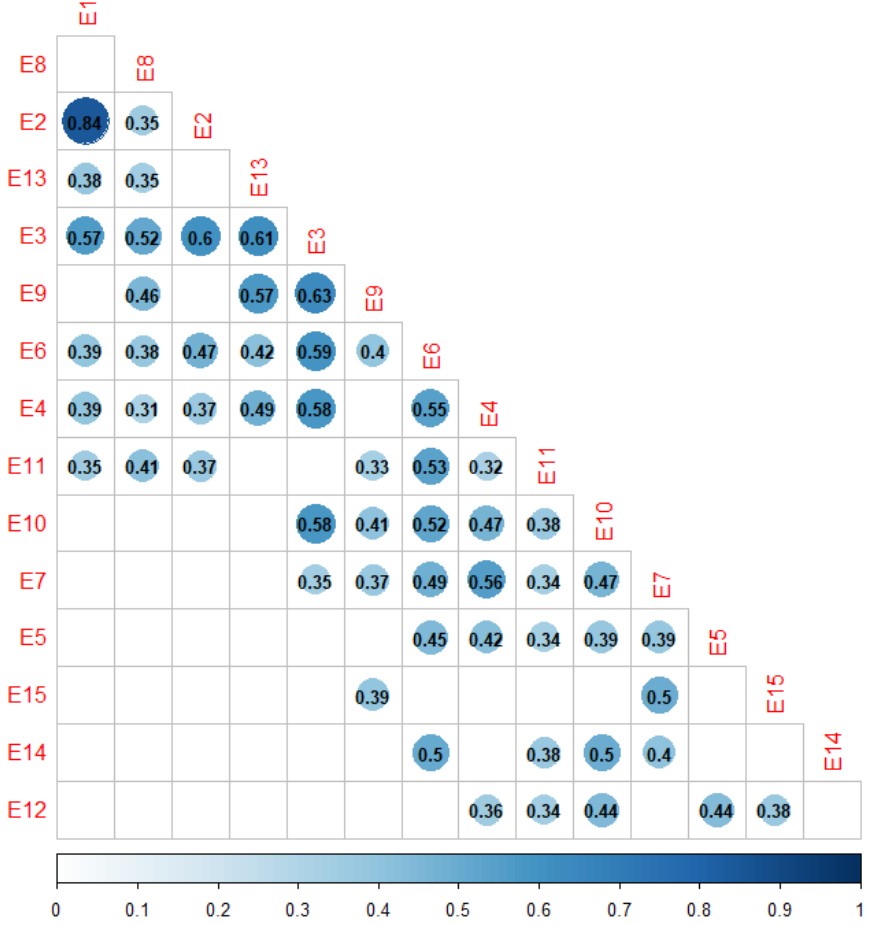

**Figure 9.** The Pearson correlation for the MOENV employees' questionnaire. The *r* values are shown only for statistically significant correlation at *p* < 0.05.

**Table 3.** Descriptive statistics from the 5-point Likert scale analysis of the MOENV employees' questionnaire.

| | Classification | Items | Mode | Mean | Std. Error of Mean | Std. Dev. |
|---|---|---|---|---|---|---|
| E1 | Awareness | Awareness of the Jordanian environmental law (E1) | 5 | 4.26 | 0.129 | 0.848 |
| E2 | Awareness | Awareness of the procedures and reports issued by the Jordanian Environment Law (E2) | 4 | 4.12 | 0.125 | 0.823 |
| E3 | Commitment | The Ministry of Environment fully cooperates with the Environment Police to implement the procedures of the law (E3) | 5 | 4.56 | 0.112 | 0.734 |
| E4 | Commitment | The Ministry explains the provisions of the law to farmers (E4) | 4 | 3.70 | 0.113 | 0.741 |
| E5 | Commitment (Farmers) | A commitment by farmers to the Jordanian Environmental Law(E5) | 3 | 2.88 | 0.134 | 0.879 |
| E6 | Commitment | A continuous supply of various environmental data to the Ministry from different regions (E6) | 4 | 3.81 | 0.121 | 0.794 |
| E7 | Commitment | The continuous coordination between the Ministry of Environment and farmers to prepare and implement environmental plans (E7) | 3 | 3.16 | 0.115 | 0.754 |
| E8 | Law enforcement | Institutions are required to submit a comprehensive study of expected environmental impacts before obtaining professional licenses (E8) | 4 | 4.19 | 0.101 | 0.664 |
| E9 | Commitment | The Ministry of Environment conducts an environmental audit for any project when it is confirmed that the activity of a particular facility causes environmental damage (E9) | 5 | 4.40 | 0.106 | 0.695 |
| E10 | Compliance (Farmers) | Farmers' obligation to disclose environmental permits related to their work or workers (E10) | 3 | 3.40 | 0.106 | 0.695 |
| E11 | Compliance (Farmers) | Commitment on the part of farmers not to use hazardous substances prohibited by law (E11) | 3 | 3.37 | 0.120 | 0.787 |

**Table 3.** *Cont.*

|  | Classification | Items | Mode | Mean | Std. Error of Mean | Std. Dev. |
|---|---|---|---|---|---|---|
| E12 | Compliance (Farmers) | Farmers' obligation to destroy and dispose of waste according to the procedures approved by the Ministry (E12) | 3 | 2.79 | 0.113 | 0.742 |
| E13 | Law enforcement | Establishments with large emissions must take the necessary measures to reduce these emissions (E13) | 5 | 4.53 | 0.090 | 0.592 |
| E14 | Farmers (commitment) | Farmers are committed to facilitating work, cooperating with the environment inspector, and submitting the necessary documents upon request (E14) | 4 | 3.58 | 0.106 | 0.698 |
| E15 | Law enforcement | The adequacy of penalties provided for the protection of the environment (E15) | 4 | 3.51 | 0.150 | 0.985 |

## 4. Discussion

### 4.1. Awareness, Commitment and Compliance

#### 4.1.1. The Farmers

The primary focus of this research was to assess the different implementation, awareness, commitment, and compliance characteristics among the study participants. Farmer compliance cannot be achieved through the first hypothetical model due to the modest commitment of farmers and lack of correlation between awareness and commitment. In addition, farmer awareness was called into question by the MOENV. This view was supported by the low compliance with environmental laws concerning the use and disposal of hazardous materials. Furthermore, the farmers' lack of awareness was not uncommon. A study on farmers' awareness of Saudi Arabia's environmental legislation attributed their low awareness to their lack of knowledge of the adverse effects of agrochemicals on the environment [40]. Both the farmer and MOENV questionnaires confirmed the farmers' weak commitment to the law. Although there is an apparent association between commitment and awareness, neither factor was correlated in the farmers' questionnaire. In fact, when it comes to farmers economic interests [38], previous findings showed a lack of correlation between awareness and commitment, and calculated motivation in the form of long-term economic benefit was more effective in improving farmers attitudes toward the environment [24] A survey carried out in northern France showed that commitment is a function of social factors rather than awareness [45]. Awareness was not correlated to compliance; however, commitment was reasonably related to compliance, at least when it comes to the farmers' consideration of the risks of environmental hazards, which is in line with previous findings that indicated a significant correlation between environmental stewardship (concerns for the environment) and attitude toward adopting AEMs [24]. Compliance in the farmers' questionnaire was contradicted by the MOENV questionnaire, which could be attributed to the lack of supervision by the MOENV, confirming previous research indicating the need to monitor the awareness and behavior of farmers to enhance their compliance [40,46], and that lack of monitoring and implementation of the rules may inhibit farmers' legal cognition, which in turn undermines farmers' awareness and commitment [30].

The second hypothetical model may be a better fit for farmers' interaction with environmental laws, but this was not confirmed due to the lack of correlation between compliance, awareness, and commitment. However, in the second model law, enforcement was the catalyst for awareness and commitment, which in turn is a product of law enforcement and follow-up from the proper authorities. Moreover, monitoring was correlated with law enforcement. However, insufficient monitoring from the MOENV prevented effective law enforcement and reflected negatively on the rest of the chain when trying to promote compliance. As previously noted, the respect to rules and regulations can only be encouraged by effective implementation of the environmental law [30].

### 4.1.2. JPMC Employees

In the JPMC questionnaire, awareness, commitment, and compliance were confirmed and reasonably correlated. Therefore, this hypothetical model is a reasonable characterization of JPMC compliance with environmental laws. It also indicates that awareness of the JPMC employees was not initiated by the management of the JPMC. Other weakness revealed by the JPMC questionnaire include the inadequacy of the penalties provided to protect the environment; they also suggest that the first model, which is based on the idea that awareness and commitment lead to compliance, is more appropriate than the second model for describing the behavior of the JPMC employees. This is because the second model depends on law enforcement to achieve compliance. Thus, this confirms that corporate adherence to environmental law is motivated by commitment rather than deterrence [37,39]. However, improving the implementation of laws may help to increase the awareness of the JPMC management, since integrated law and regulation enhances the mining industry's compliance with the environmental laws [47]. Additionally, recent reviews of the environmental impacts of mining operations emphasized the importance of allocating more resources to environmental law enforcement [48,49].

### 4.1.3. The MOENV

As previously indicated, the farmers showed poor legal compliance due to lax monitoring and inadequate follow-up by the MOENV. Furthermore, moderate correlations were found between awareness and commitment and between commitment and law enforcement. This suggests that the MOENV needs to expand its monitoring activities. Enhanced implementation of the law may enhance the environmental quality and economic benefits of the phosphate mining industry [50].

## 5. Conclusions

### 5.1. Summary

This research is considered the first study to evaluate the effectiveness of the Jordanian environmental laws. This study considered the agricultural lands near phosphate mines, and thus, the following three stakeholders were involved: the MOENV, the JPMC employees, and the farmers near the phosphate mines. Compliance was studied in relation to the following four variables: (1) the monitoring of environmental violations, (2) law enforcement in the form of fines and penalties, (3) commitment that covers issues of moral values and concern for the environment, and (4) awareness about the environmental laws. Two hypothetical models were considered to study the role of variable compliance. The first model was based on awareness, which subsequently fosters commitment and then leads to compliance. The second model was based on monitoring leading to enforcement, and enforcement was mediated by commitment and awareness, with both variables contributing to compliance. The results highlighted the following issues:

- The first model explained the compliance by the JPMC employees, which was confirmed by the strong correlation of awareness with commitment as well as commitment with compliance. Thus, the JPMC's adherence to environmental laws was driven by the awareness and commitment of its employees.

- It was found that the management at the JPMC should increase its involvement in enhancing the awareness of its employees regarding the environmental laws.
- The law enforcement items, including the effectiveness of the penalties prescribed by the environmental laws, were questioned by the JPMC employees, which supports the results of the first model, suggesting that the JPMC's compliance is based on commitment and normative motivation.
- The first model cannot be applied to explain the farmers' compliance due to modest compliance from the farmers and their low awareness and modest commitment. Additionally, this is further supported by the lack of correlation between awareness and commitment.
- Because monitoring was correlated with law enforcement in the farmer's questionnaire, the farmers' lack of compliance maybe attributed to the inadequate monitoring of the MOENV and law enforcement.
- The farmers' commitment was correlated with compliance.

### 5.2. Approaches to Compliance with the Environmental Law

The findings of this research indicated that both farmers and mining corporates like the JPMC can comply with the environmental laws, however, through two different approaches. Therefore, the effectiveness of the environment hinges on understanding the factors that lead to compliance. For JPMC, compliance is driven by normative variables such as commitment and corporate image, which was sufficient to ensure JPMC compliance even with the absence of enforcement or when the penalties did not offer sufficient deterrence; however, effective monitoring can motivate the management of mining corporations to enhance their awareness programs. Regarding farmers, commitment is an essential variable to ensure their compliance with the environmental law, but it is not sufficient since farmers may have positive personal perspectives toward the environment; however, their attitudes may differ when it comes to their business economic benefits. Therefore, calculated motivation in the form of effective monitoring and enforcement is essential to enhance compliance. It should be noted that enforcement does not replace commitment; rather, it integrates with awareness and commitment.

The results of this research have local and global application since they demonstrate the importance of stewardship to achieve compliance with environmental laws and regulations. This was evident for a corporation like the JPMC, while for farmer commitment, good stewardship was the key to compliance; however, farmers' compliance can be improved by paying more attention to the farmers' economic interests and effective monitoring and implementation of the rules and regulations.

### 5.3. Recommedations

The following recommendations are presented in order to increase the effectiveness of environmental laws:

- More training programs are recommended for both the employees and managers of regulated entities in order to achieve a better understanding of the importance of the environment.
- A detailed study into the factors influencing farmers' awareness and commitment to environmental laws is needed as a means of increasing farmers' compliance with the law.
- Allocation of adequate resources to the regulating agencies in order to increase the monitoring law enforcement activities.

**Author Contributions:** Conceptualization, M.R.; Methodology, A.B.; Investigation, A.B.; Data curation, A.B.; Writing—original draft, M.R.; Writing—review & editing, M.R.; Supervision, M.R. All authors have read and agreed to the published version of the manuscript.

**Funding:** This research received no external funding.

**Institutional Review Board Statement:** The research was done per the rules and regulations of the University of Jordan. The research methodology, including the questionnaires, was reviewed, and approved on 20 December 2022, by the Council of Graduate Studies.

**Informed Consent Statement:** All the respondents were aware that the questionnaires were for research purposes. The administrations of the MONEV and JPMC approved the questionnaires and allowed the participation of their employees.

**Data Availability Statement:** The data can be obtained from the corresponding author upon request.

**Conflicts of Interest:** The authors declare no conflict of interest.

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
