# Peer review of "The Role of the Law in Prompting Environmental Stewardship for Farms Located Near Phosphate Mines"

_sustainability, doi:10.3390/su16031140_

Round 1
Reviewer 1 Report
Comments and Suggestions for Authors
The authors investigate the effectiveness of the Jordanian environmental law in protecting agricultural lands near phosphate mines operated. Related actions such as allocating more resources to increase the monitoring activities and studying the social and economic factors influencing farmers' compliance are recommended. The study is interesting. Further comments are listed below:
1) The picture needs to be improved. For example Pictures 1,2,3, too sample and need to be modified to contained more information or reduced the size of the picture.
2) The research gap and the significance of your work should be clearly clarified in the last paragraph of the introduction section,
3) The conclusion and the contribution should be summarized and tighten conclusion parts.
4) The English is needed to be checked and improved. The “.” is missing in the last sentence.
5) The study focuses on the issues in Jordanian environmental law, but for research purpose, how it can be used in other areas and what is the new universal idea that can be used for other countries, not only in Jordanian.
Comments on the Quality of English Languageneeded to be improved
Author Response
File attached

Reviewer 2 Report
Comments and Suggestions for Authors
This article investigates the environmental management issues of farms located near phosphate mines. The author points out that phosphate mining is one of the main environmental problems in Jordan, and mining areas have caused serious environmental impacts. The author believes that the implementation of environmental law requires the cooperation of three stakeholders, namely law enforcers, environmental drivers, and beneficiaries. The main conclusion of this article is that environmental law plays an important role in promoting farm environmental management. Environmental laws can encourage farmers to take environmental measures and reduce negative impacts on the environment. In addition, environmental laws can also increase public awareness of environmental issues and promote social attention to environmental protection issues. This article is helpful to scholars in related fields. But before publishing this article, please respond to the following questions.
1. In line 124-130, the author considers the implementation of environmental law to rely on three factors: enforcers, environmental drivers, and beneficiaries. What is an environmental driver? This article does not provide a clear and detailed explanation of this concept.
2. What is the basis for the author's reliance on three factors for the implementation of environmental law, namely law enforcers, environmental drivers, and beneficiaries? Has anyone ever proposed such a theoretical model before? If so, please provide relevant literature. The validity of this theory cannot be based on one's own assumptions and should be supported by a large amount of previous research and data.
3. The author relies on three factors for the implementation of environmental law, namely enforcers, environmental drivers, and beneficiaries. These factors all belong to human factors. The implementation of any law cannot be separated from natural environmental factors, why does this theory not include the impact of natural environment? Considering only human factors is not sufficient, and the impact of natural factors and environmental conditions on the difficulty of implementing environmental laws should also be considered.
Is the theoretical model proposed in Figure 2 supported by evidence? Is there any literature support?
Figure 2 A theoretical model describing a confidential relationship between awareness, commit -146
Ment and compliance
5. Is the theoretical model proposed in Figure 3 supported by evidence? Is there any literature support? Why can monitoring generate enforcement? Is the conclusion that combining Commitment and Awareness can generate compliance too absolute?
6. Many sentences in the article do not align with grammar and logic. For example: lines 116-120, "The objective of this study is in view of the environmental challenges Jordan is fixing and the series of hazards of the photographic industry it is important to evaluate the effect."
Comments on the Quality of English Languageneed to be improved
Author Response
File attached

Reviewer 3 Report
Comments and Suggestions for Authors
In this study, the evaluation of the effectiveness of the environmental law was based on a hypothetical model that considers consecutive relations between awareness, commitment, and compliance. However, authors do not justify why these variables in the study are used. They only pointed out "Furthermore, environmental awareness is growing at a high level in Jordan as upholding a safe and secure environment is seen as a human right." Moreover, it seems there are prior studies when authors said "the perception toward the law is characterized awareness, and commitment as well as awareness and commitment lead to compliance”. So, it must argue and cite about this issue.
Authors also uphold that the implementation of the environmental law depends on the interaction between three groups of stakeholders. However, sample is poor because it is formed by 43 of employees in the Jordanian Ministry of Environment, 36 workers of the Jordan phosphate Mines Company and 35 farmers. In fact, reliability and descriptive statistics are too weak in order to explain the great impact of law in prompting environmental stewardship for farms located near phosphate mines, which is authors' purpose. Consequently, discurssion and conclusion of this manuscript only should glimpse a previous statement which requires a follow research.
Author Response
File attached

Round 2
Reviewer 1 Report
Comments and Suggestions for Authors
The authors revised the manuscript accorrdingly, and no further comments is applied.
Comments on the Quality of English LanguageEnglish is ok.
Author Response
Thank you very much for your insights.

Reviewer 2 Report
Comments and Suggestions for Authors
The author responded to my question and I believe that this article can now be published
Author Response
Thank you very much for your valuable insights.
Reviewer 3 Report
Comments and Suggestions for Authors
In introduction section, it's necessary to point out lack of the prior research of the effectiveness of environmental law and the model proposed. Authors must explain a statement of the fronteir of knowledge about this issue in order to indicate the contributions of the manuscript. The two models proposed must be based on other previous research and its findings in other countries.
Carefully, the section "2.3. Theory" must be changed with a different tittle like 'the two proposal models', 'two theorical models' or anyway similar title. Otherwise, it is plenty confusing.
In discursion and/or conclusion section, you also must report the contributions of this study in relationship with literature. For example, I can read relations/correlations of both models at pag 19 like 'findings', but it is not the contributions of your research toward the literature as well as it is not exist a link with the fronteir of knowledge.
Author Response
Thank you very much for your valuable insights, File attached

Round 3
Reviewer 3 Report
Comments and Suggestions for Authors
Authors have improved the manuscript. However, there are several points that should be enhaced.
In introduction section, you must indicate the contributions of the manuscript about the two models proposed must be based on other previous research and its findings.
In discursion and/or conclusion section, you also must include the theoretical and practical implications to improve the understanding of the readers, instead 'correlations' as findings.
